# Spatial Heterogeneity and Temporal Variation in Urban Surface Albedo Detected by High-Resolution Satellite Data

Hantian Wu [1], Bo Huang [1,*], Zhaoju Zheng [2], Zonghan Ma [2] and Yuan Zeng [2,3]

1   Institute of Space and Earth Information Science, Chinese University of Hong Kong, Shatin, Hong Kong, China
2   State Key Laboratory of Remote Sensing Science, Aerospace Information Research Institute, Chinese Academy of Sciences, Beijing 100101, China
3   School of Resource and Environment, University of Chinese Academy of Sciences, Beijing 100049, China
*   Correspondence: bohuang@cuhk.edu.hk

**Abstract:** Albedo is one of the key parameters in the surface energy balance and it has been altered due to urban expansion, which has significant impacts on local and regional climate. Many previous studies have demonstrated that changes in the urban surface albedo are strongly related to the city's heterogeneity and have significant spatial-temporal characteristics but fail to address the albedo of the urban surface as a unique variable in urban thermal environment research. This study selects Beijing as the experimental area for exploring the spatial-temporal characteristics of the urban surface albedo and the albedo's uniqueness in environmental research on urban spaces. Our results show that the urban surface albedo at high spatial resolution can better represent the urban spatial heterogeneity, seasonal variation, building canyon, and pixel adjacency effects. Urban surface albedo is associated with building density and height, land surface temperature (LST), and fractional vegetation cover (FVC). Furthermore, albedo can reflect livability and environmental rating due to the variances of building materials and architectural formats in the urban development. Hence, we argue that the albedo of the urban surface can be considered as a unique variable for improving the acknowledgment of the urban environment and human livability with wider application in urban environmental research.

**Keywords:** urban surface albedo; spatial heterogeneity; seasonal variation; Sentinel-2

## 1. Introduction

Urban land cover has been expanding rapidly since the middle of the 20th century. The urban land in Southeast Asia increased six times from 1992 to 2018 [1]. By 2030, the urban land cover will expand to thrice that in 2010 [2]. Furthermore, another 2.5 billion people are projected to inhabit urban areas worldwide by 2050 [3]. These changes will place extreme pressure on the urban environment through various aspects including the thermal environment and livability. Increasing impervious cover expansion has occurred universally in cities [4]. The urban heat island (UHI) is one of the most remarkable effects of urban expansion and has a wide-ranging association with local weather patterns and plant phenology [5,6]. The amount of solar radiation reflected by the Earth's surface increases because of the light building materials introduced during urbanization and the decreased vegetation coverage. There will be a series of effects, such as increased risks to human health, reduced livability of the environment [7], unpredictable ecological effects, and climate disasters [8]. The expansion of urban land and the continuing effects of global climate change will further expand the UHI region, and the number of people affected by extreme temperatures [9].

The effect of urbanization on global climate change has two parts: the intensity of human activities and the modification of the geometry and composition of surface elements [10–12]. There is no clear separation between these two parts because both increased

with urbanization. However, urban surface albedo variation (which is the ratio of short-wave upward radiation from the ground surfaces to the downward radiation of sunshine) is one of the major effects of urban geometry surface component variation [4,13–17]. The UHI and other localized climate changes are the direct impacts of local albedo variation of the urban surface.

Urban surface albedo contributes to the UHI, including the absorption of shortwave radiation, longwave radiation reduction, and turbulent heat transportation losses [18]. Hence, albedo has been used as a dependent variable to contribute to UHIs and land surface temperature (LST) in previous urban thermal studies [19–22]. Urban canyons can absorb solar radiation [23], which is a typical indicator that contributes to UHIs [24]. In fact, the factors affecting albedo are not the same as those affecting UHI and LST but are more related to material and geometry. On the one hand, urbanization changes the geometric composition of urban areas and street areas. Impervious surfaces alter the ground surface in multiple ways affecting UHIs. The changes in topography associated with urbanization contribute to changes in the albedo of the urban surface by replacing natural vegetation or bare ground with man-made materials [10,25–27]. Various studies [19,21,28–32] have stated that applying light building materials and changing the layout of urban areas increase the surface albedo of urban areas and human settlements [2,33]. However, urbanization may have contrasting effects on the albedo of the urban surface. The urban geometry will influence the thermal environment of urban areas by blocking solar radiation, and the blocked solar radiation will cross-reflect and be absorbed by the building surfaces [34]. The replacement of vegetation elements by streets, building facades, and roofs can reduce the albedo of the urban surface [26,35,36]. In this regard, the greater the building density is, the more radiation is trapped, reducing the overall albedo of the urban surface and the magnitude of its reflectivity.

Recently, urban surface albedo has been assessed with satellite products [10,16,17,37–39]. Using remote sensing-based albedo to assess the urban energy budget and modifying the albedo of building surfaces to mitigate near-surface temperatures has become a practical option [26]. For example, satellite albedo products have been used to estimate the effects of urban land changes on radiation [16,17,40–42]. Several studies have analyzed the relationship between albedo and impervious expansion using 500 m resolution data [6,10,43,44]. Other research applied 30 m resolution satellite data to investigate the albedo changes caused by urbanization [26,40]. However, these studies did not analyze the albedo heterogeneity among different urban regions. A deeper understanding of the urban surface albedo with urban heterogeneity, such as topographic and material effects, requires high-resolution data. Many other factors, such as latitude, zenith angle, building density and height, and seasonal variation can affect the urban surface albedo. Previous researchers have focused on studying the annual variation in albedo [2,17,26,39,40], but the seasonal variation associated with the albedo of the urban surface has been less studied [26,27].

The urban surface albedo is heavily dependent on the building density and height, which are related to the urban development process [40]. However, urbanization in China has been rapid and continuous since the reform and opening-up of China. The synchronous development of different cities may not be a good case for highlighting the influence of different development stages on albedo variation because the buildings of cities applied similar materials, landscapes, and urban layouts. Exploring the relationships between albedo and urban development stages requires associating albedo in internal and external urban areas with human livability and the urban environment, which are the central guiding principles for decision-makers in urban development planning [45]. Livability is an all-encompassing term that relates to sustainability, biodiversity, and ecosystems [46,47]. Urban livability and environmental ratings emphasize that urban development should balance impervious surfaces and green cover [48,49] as a proxy for development stages. Hence, associating albedo with urban livability and environmental rating is a potential option for improving the understanding of the relationship between albedo and urban development.

This study seeks to explore the spatial-temporal characteristics of urban surface albedo, and whether it can be a unique indicator in thermal environmental research on urban spaces. The first goal is to detect urban heterogeneity with high-resolution albedo data. The second goal is to understand the seasonal urban heterogeneity with albedo. The third task is exploring the association between the albedo and indicators such as fractional vegetation cover (FVC) and land surface temperature (LST). The last goal is to address the relationship between albedo and urban development stages using proxies of livability and environmental rating.

## 2. Materials and Methods

### 2.1. Study Area

According to the global livability ranking and environmental rating, Bangkok, Beijing, Hong Kong, and Tokyo are selected as the study areas. An urban development stage is usually characterized by pollution, development, and renovation, so the livability ranking, and environmental rating are often highly correlated with the development stages. The selection of cities with different levels of livability will enable us to study the impact of different stages of development on albedo.

Bangkok (central coordinates: 13.7°N, 100.5°E) is the capital and the largest city of Thailand. During the 1960s and the 1990s, Bangkok experienced rapid growth and development. As a result of poor urban planning, the cityscape has become haphazard, and the infrastructure is insufficient to handle the city's growth.

Beijing (central coordinates: 39.9°N, 116.4°E) is the second largest city in China after Shanghai and is the capital of the People's Republic of China. It is one of the oldest cities in the world, with a history of over three millennia. Over the past 50 years, it has experienced several stages of development, resulting in a mixed cityscape that combines ancient and modern elements. In recent decades, Beijing has undergone extensive renovations to prepare for hosting the 2008 and 2022 Olympics, which have modernized this ancient city.

Hong Kong (central coordinates: 22.3°N, 114.1°E) is a city and a special administrative region of China that lies on the eastern shore of the Pearl River Delta. Hong Kong's urban development can be divided into two periods: before and after the Second World War. During the 1920s, when Hong Kong's population was limited, its urban development was concentrated in the nearby areas of Kowloon and Victoria Bay. Intensified immigration after the Second World War necessitated the planning and development of satellite towns in Hong Kong after the war. Since the 1950s, various satellite towns have been planned and developed.

Tokyo (central coordinates: 35.6°N, 139.6°E) is Japan's capital and largest city. The urbanization of modern Tokyo began in the 1950s and lasted until the end of Japan's economic bubble in the 1990s. Even though the expansion of Tokyo is almost over, its redevelopment is continuing.

### 2.2. Data

#### 2.2.1. Satellite Data

The Sentinel-2 mission consists of two identical satellites (Sentinel-2A and Sentinel-2B) in the same sun-synchronous orbit at a mean altitude of 786 km, with high revisit times of two to three days at mid-latitudes. Sentinel-2A was launched on 23 June 2015, and Sentinel-2B was launched on 7 March 2017. The Sentinel-2 series satellites have 13 bands with spatial resolutions between 10 m and 60 m. This research applied the Sentinel-2 MSI: Level-2A Surface reflects data product which has atmospherically corrected.

Landsat series images have been widely used for analyzing albedo variations with urban expansion due to their long-term data availability and similar spectral characteristics [6,10,26,40,43]. Landsat 8 was launched on 11 February 2013. This satellite carries the Operational Land Imager (OLI) and the thermal infrared sensor (TIRS) instruments collecting multispectral images with a resolution of 30 m (panchromatic band 15 m) and thermal

infrared images with a resolution of 100 m. This research selected the Landsat 8 OLI Level 2, Collection 2, Tier 1 SR Reflectance Composite, which has been atmospherically corrected.

All the satellite data (Table 1) have been processed for the cloud-removal task, which cannot remove one hundred percent of all clouds and shadows. Therefore, the datasets with minor cloud effects are selected.

**Table 1.** Dates of Sentinel-2 and Landsat-8 data.

|  | **Spring** | **Summer** | **Autumn** | **Winter** |
|---|---|---|---|---|
| Sentinel-2 | 3 March 2020 | 14 June 2019 | 19 September 2019 | 14 December 2018 |
| Landsat-8 |  |  |  | 4 December 2018 |

### 2.2.2. Urban Block Data

The heterogeneity of albedo and canyon effects is related to the urban layout and the height of buildings. The data contain 141,375 street blocks in 63 Chinese cities in 2017 in the format of GIS Shapefiles based on the Open Street Map database.

In this research, the floor data are clustered into low, medium, and high floor classes based on building volume. The People's Republic of China's urban and rural planning laws in 2015 provided a general description of the building volume. However, there is no specific guidance for further classification concerning the number of floors. We select the customary description of floor area ratio (FAR), where a FAR smaller than 1 is the ground floor area, a FAR in the range of 1 to 2 is a low floor of a building, a FAR in the range of 2 to 4 is a medium floor of a building. A FAR over 4 indicates a high floor of a building.

### 2.2.3. Sampling Data

Setting a linear sampling aims to demonstrate the variation in albedo across urban areas to understand the heterogeneity of the albedo in the city. To characterize the spatial heterogeneity of the albedo, the sampling grids are composed of hundreds of 10 m × 10 m rectangular cells over a 0.5 km × 50 km region in the Sentinel-2 Level 2A 10 m data (Figure 1). The size of this grid is chosen to set a uniform sampling size. The expected output of the sampling method is that each cell can contain albedo and location heterogeneity. In addition, this research also applied 1000 points that are random samples throughout the city for integration analysis with other data to be compatible with LST, FVC, and urban block data.

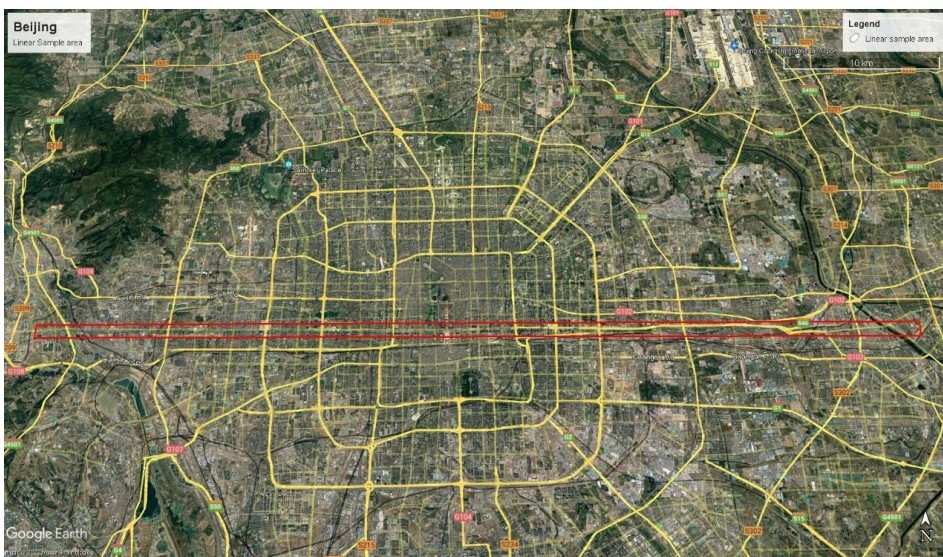

**Figure 1.** Sampling line across the city from east to west in Beijing: Captured from Google Earth Pro.

2.2.4. Livability and Environmental Rating

The Global Livability Index is compiled by the Economist Intelligence Unit (EIU) and the research and analysis division of the Economist Group. More than 30 quantitative and qualitative factors are considered in the index, covering 173 cities worldwide [50]. All cities were rated based on livability scores across five broad categories: stability, healthcare, culture and environment, education, and infrastructure. The livability and environmental ratings of the Global Livability Index 2018 were used in this research [50].

*2.3. Methods*

2.3.1. Albedo Estimation and Spatial-Temporal Variation Analysis

Research on the land surface temperature environment requires broadband shortwave albedo [51]. Both Sentinel-2 MSI and Landsat 8 OLI datasets are provided as multiple bands with narrow spectral ranges. To calculate the shortwave broadband albedo from Landsat data, narrow-to-broadband conversion coefficients were derived from laboratory spectra [38]. Both Wang [52] and Bonafoni [41] followed the strategy of using the conversion coefficients to derive the laboratory spectral to the broadband shortwave albedo and developed narrow-to-broadband albedo conversion coefficients for Landsat-8 and Sentinel-2 separately:

$$a = \sum_{B=1}^{N} (p_b \times w_b) + C \tag{1}$$

where $p_b$ is the surface reflectance for a specific Sentinel-2 MSI and Landsat 8 OLI bands b. $w_b$ is the conversion coefficient, and C is the constant provided by Bonafoni [41] and Wang [53] in Table 2:

**Table 2.** Weighting coefficient ($w_b$) for albedo.

|  | $w_2$ | $w_3$ | $w_4$ | $w_5$ | $w_6$ | $w_7$ | $w_8$ | $w_{11}$ | $w_{12}$ | C |
|---|---|---|---|---|---|---|---|---|---|---|
| Sentinel-2 | 0.2266 | 0.1236 | 0.1573 |  |  |  | 0.3417 | 0.117 | 0.0338 | 0.0 |
| Landsat-8 | 0.2453 | 0.0508 | 0.1804 | 0.3081 | 0.1332 | 0.0521 |  |  |  | 0.0011 |

2.3.2. Spatial Heterogeneity and Entropy, Temporal Variation

Albedo spatial pattern differences can be analyzed by comparing the albedo data derived from Sentinel-2 and Landsat-8 at different resolutions. The analysis process of spatial distribution characteristics and differences in albedo requires samples for three typical urban landscapes of business, height-density, and urban–rural mixed areas, along a sampling line. Further, the change in albedo with season will be analyzed based on the comparison of albedo in four seasons along the sampling line and typical urban landscapes.

A rigorous method for measuring spatial heterogeneity in urban sprawl is entropy [54], which is widely used to integrate remote sensing and GIS [55–57]. Entropy measures can also be proposed and computed in situations such as remote sensing with limited information but extensive availability. Shannon's entropy formula [58]:

$$\text{Entropy} = \sum_{i=1}^{I} p(x_i) \times \log\left(\frac{1}{p(x_i)}\right) \tag{2}$$

where the I is the total pixel and i is the ith pixel. $p(x_i)$ is the probability of the ith outcome, and $\log\left(\frac{1}{p(x_i)}\right)$ is the measurement of information from $x_i$.

2.3.3. FVC

Fractional vegetation cover (FVC) is defined as the ratio of vegetation cover area to total land cover area [59,60]. The FVC not only reflects the size of the plant's photosynthetic area and the density of vegetation growth but also indicates the growing trend of vegetation

to a certain degree [61]. Therefore, this parameter is defined as an indicator to evaluate the status and development of terrestrial ecosystems [62]. FVC has many applications, including soil loss [60] and urbanization expansion estimation [63], among others. The Satellite data on the same date as albedo are used to estimate FVC and investigate the relationship between FVC and albedo. The FVC formula is shown below:

$$FVC = \frac{NDVI - NDVI_s}{NDVI_v + NDVI_s} \qquad (3)$$

where NDVI is the normalized difference vegetation index; $NDVI_s$ is the vegetation index of bare soil pixel; and $NDVI_v$ is the vegetation index of the most vegetated areas.

### 2.3.4. Land Surface Temperature (LST)

LST is another indicator used to estimate UHIs [44,64,65]. LST retrieval relies on the thermal sensors of the satellite. The satellite data at the same date as albedo are used to estimate LST and investigate the relationship between FVC and albedo. For the Landsat series satellite, the USGS provides a group of formulas for estimating LST. First, it is necessary to calculate the spectral radiance at the top of the atmosphere (TOA):

$$TOA = M_L \times Q_{cal} + A_L \qquad (4)$$

where $M_L$ is the band-specific multiplicative rescaling factor from the metadata. $Q_{cal}$ corresponds to band 10, which is the thermal infrared band. $A_L$ is a band-specific additive rescaling factor from the metadata.

Then, the TOA is used to calculate the brightness temperature (BT):

$$BT = (K2/(\ln(K1/TOA) + 1)) - 273.15 \qquad (5)$$

where the K1 and K2 are the conversion constants from Landsat-8 band 10.

The next step is calculating the emissivity ($\varepsilon$) from the FVC:

$$\varepsilon = 0.004 \times FVC + 0.986 \qquad (6)$$

The last step is calculating the LST:

$$LST = (BT/(1 + (0.00115 \times BT/1.4388) \times Ln(\varepsilon))) \qquad (7)$$

### 2.3.5. Statistical Analysis and Graphical Analyses

Linear correlations and box plots were conducted using Microsoft Excel and OriginLab to assess the association between the albedo and indicators of building heights, urban densities, LST, FVC, livability and environmental rating. The standard $p < 0.01$ was used as the confidence level for statistical significance.

## 3. Results

### 3.1. Spatial Heterogeneity of Albedo under Different Resolutions

The overall pattern of albedo distribution in Beijing is low in the city center and high in the surrounding rural–urban mixed areas (Figure 2). The business area is in the east of the central city, which is full of skyscrapers and generally has low albedo. The old building area has an extremely high building density on the north and south sides of the central city area, showing a relatively higher albedo. The new urban areas have expanded outside of the old building area in recent decades and the outside of the city area is the rural–urban mixed area, which shows much higher albedo than that in the city center. The ranges of spatial entropy values are listed in Table 3, which indicate that the increase in entropy has a significant association with the resolution of satellite data.

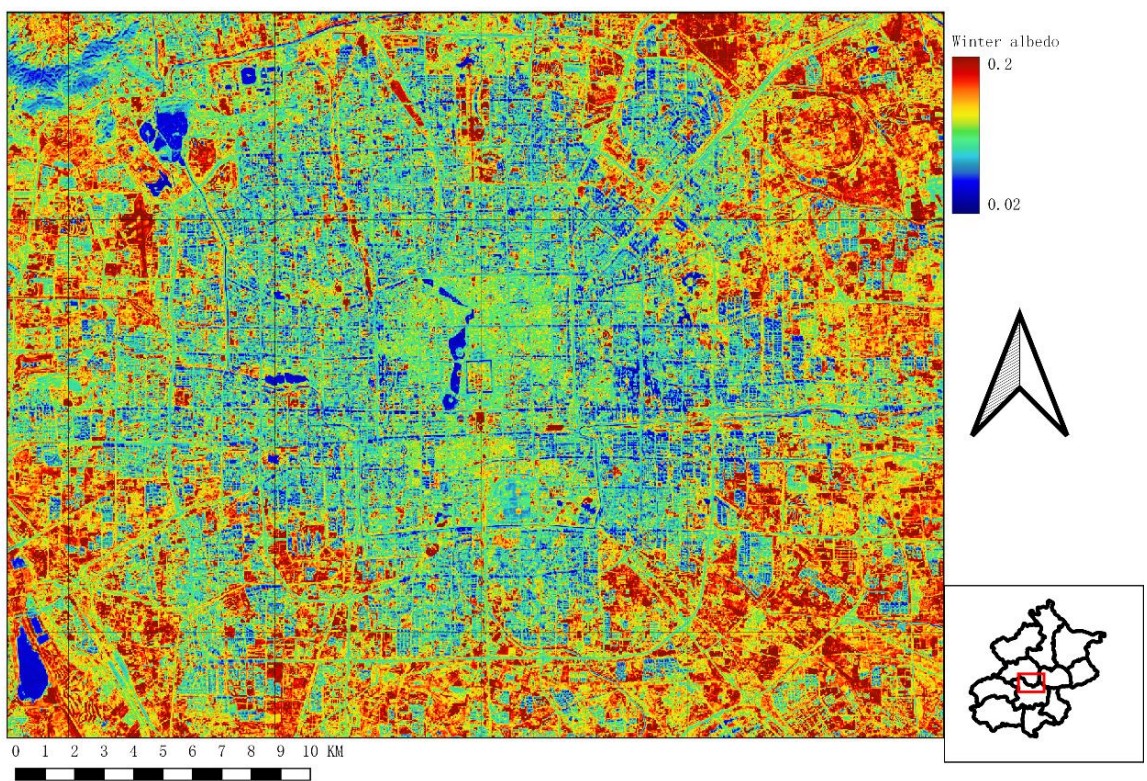

**Figure 2.** Albedo derived from Sentinel-2 in Beijing at winter (14 December 2018).

**Table 3.** Entropy of albedo derived from Sentinel-2 and Landsat-8 in different seasons.

|  | Spring | | | Summer | | | Autumn | | | Winter | | |
|---|---|---|---|---|---|---|---|---|---|---|---|---|
|  | Sentinel | Landsat | (%) | Sentinel | Landsat | (%) | Sentinel | Landsat | (%) | Sentinel | Landsat | (%) |
| Business | 0.0497 | 0.0362 | 37.3 | 0.0449 | 0.0408 | 10.1 | 0.0476 | 0.0435 | 9.6 | 0.0410 | 0.0379 | 8.1 |
| High-density | 0.0503 | 0.0365 | 38.1 | 0.0448 | 0.0397 | 12.8 | 0.0473 | 0.0424 | 11.6 | 0.0448 | 0.0413 | 8.5 |
| Rural | 0.0382 | 0.0351 | 8.8 | 0.0377 | 0.0342 | 10.2 | 0.0376 | 0.0351 | 7.2 | 0.0409 | 0.0334 | 22.4 |

Figure 3 demonstrates the low albedo distribution characteristics of the central business area, including the skyscrapers, resident regions, green spaces, main roads, and streets. The skyscrapers in a small region block the light, decrease the surrounding region albedo and create a region where albedo is close to 0.02 (e.g., Figure 3 red box area). The difference in street and the building albedo is clear in the medium-height residential area (e.g., Figure 3 black box area). The street albedo is significantly lower than that of the main road and is not affected by the shadow of buildings. In Figure 3B, the low albedo value area (red box) is smaller than that in (Figure 3A), which benefits from the resolution of the Sentinel-2 data. The albedo of Sentinel-2 can demonstrate the contrasts of albedo between buildings and street yards, while Landsat-8 albedo is hard to represent, especially in the black box region where the canyon effect works on the street surfaces from the height variation.

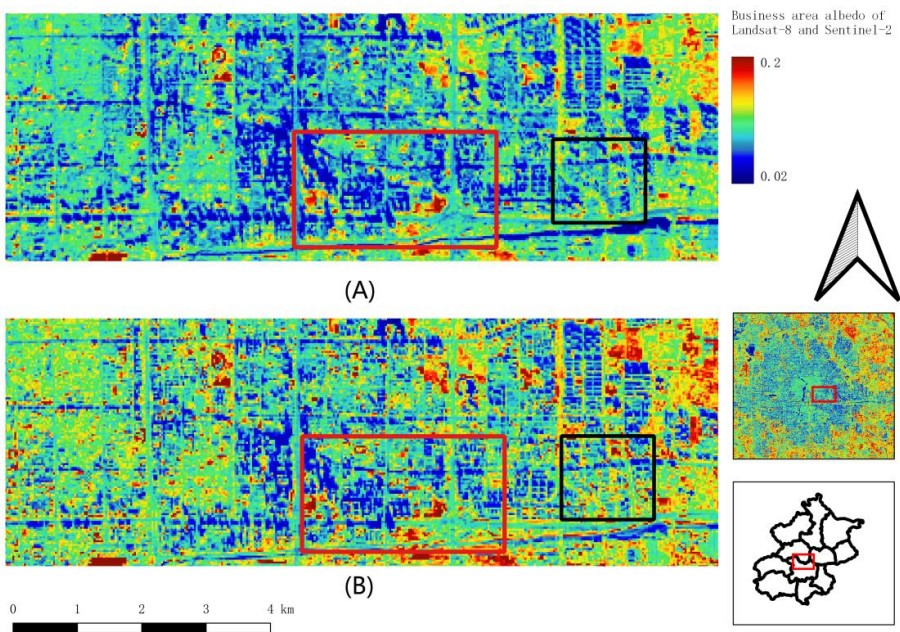

**Figure 3.** Albedo of the business area in winter, the black box: the urban residential area, and the red box: business area. (**A**) Landsat, 4 December 2018; (**B**) Sentinel-2, 14 December.

The albedo maps of high-density areas, including old urban residential areas, green covers, main roads, and streets are shown in Figure 4. The main characteristic of this area is that the albedo is evenly distributed among the blocks of the area. The buildings in the yellow box area were built over 70 years with narrow streets. The contrast in albedo between the streets and residential blocks is difficult to identify from the Landsat-derived albedo map (Figure 4A), because the streets may only be 3 m wide and the distance between the buildings in the ranges from 0.5 m to 3 m. Figure 4B is the Sentinel-2 albedo of the high-density area which has more details of albedo variances of buildings.

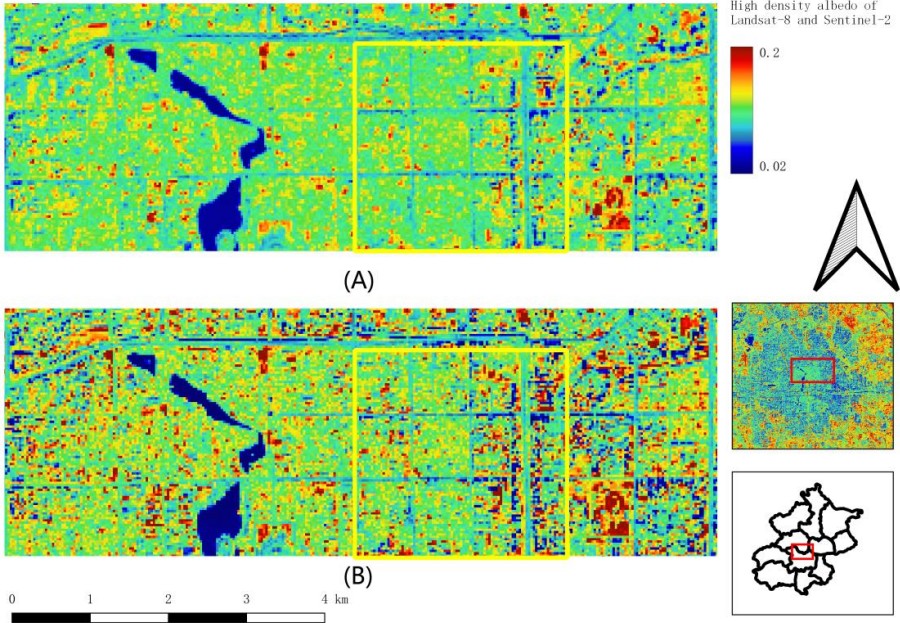

**Figure 4.** Albedo of high-density area in winter, the yellow box: residential area. (**A**) Landsat, 4 December 2018; (**B**) Sentinel-2, 14 December 2018.

Figure 5 displays the albedo characteristics of the urban–rural mixed areas, including urban residential buildings, rural residential buildings, green spaces, agriculture and bare land, main roads, and streets. There are various characteristics in this region: the first one is the area full of urban residential buildings, which shows that the street and building albedo is between 0.1 and 0.02 (Figure 5 black box). Figure 5 yellow box is residential areas similar to the yellow box in Figure 4. However, the street albedo is higher than the building albedo due to the lower building density. This area's albedos derived from different sensors are much different; the albedo of Landsat-8 is approximately 0.1 (Figure 5A) and the albedo of Sentinel-2 is approximately 0.2 (Figure 5B). The purple box contains rural buildings and the agricultural land has a high albedo (e.g., over 0.2).

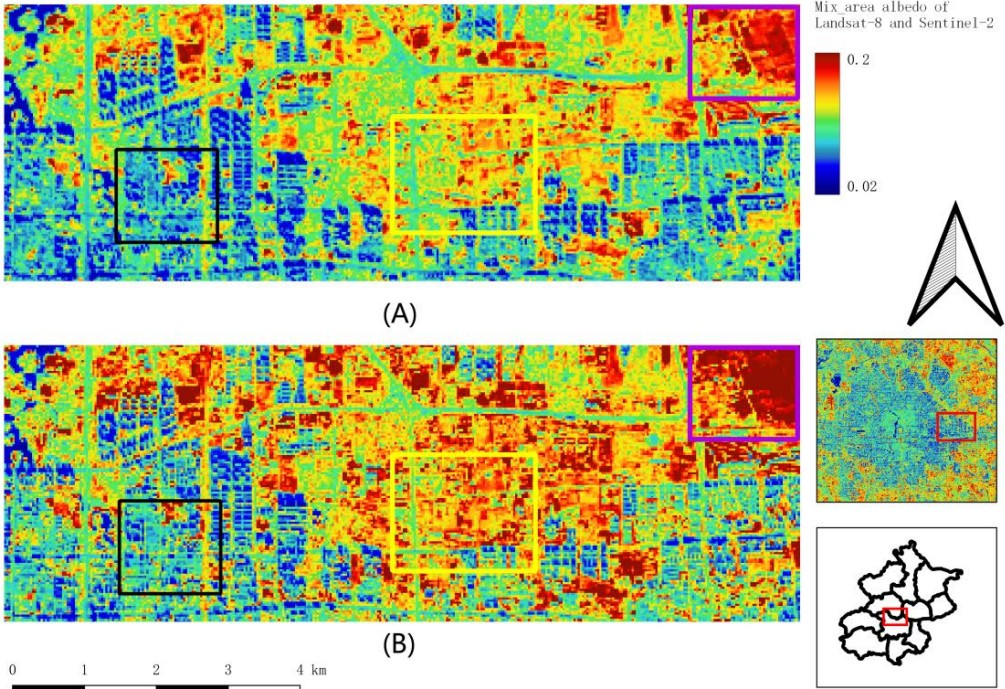

**Figure 5.** Albedo of mixed area in winter, black box: urban residential area, yellow box: rural residential area, purple box: agriculture and bare soil area. (**A**) Landsat, 4 December 2018; (**B**) Sentinel-2, 14 December 2018.

Figure 6 displays Landsat-8 and Sentinel-2's average albedo of four seasons along the sampling line in Beijing, which covers the three major research regions: business, high density, and mixed area. Figure 6 shows that Sentinel-2's mean albedo is higher than Landsat-8's mean albedo. The square is an impervious surface but a high albedo region which suggests that a large impervious surface can also have similar albedo reflection characteristics as rural areas in both Landsat-8 and Sentinel-2 satellites in a certain spatial layout. The main contrasts between albedos derived from both Sentinel-2 and Landsat-8 are in the black, red, and yellow boxes which are full of skyscrapers and medium–high floor buildings. In this area, the albedo of Sentinel-2 has more details in the narrow streets between buildings than the albedo from Landsat-8.

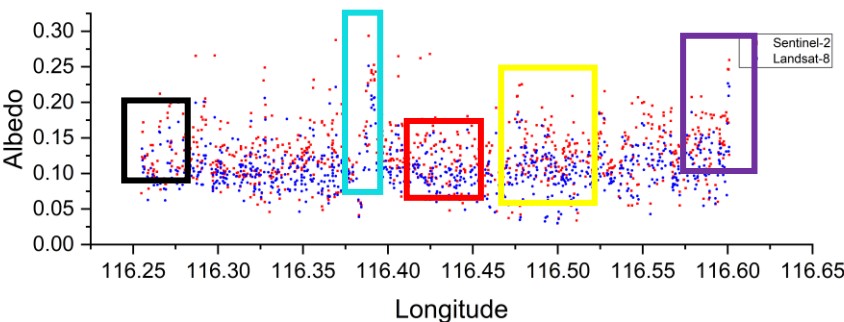

**Figure 6.** Average albedo of four seasons along the sampling line: the black box: the urban residential area, the red box: business area, yellow box: rural residential area, the purple box: the agriculture and bare soil area, the blue box: regulatory square.

### 3.2. Seasonal Variation of Albedo

Beijing has four distinct seasons with distinct albedo values in each. Due to clouds, precipitation, or other climatic effects, spring and autumn may have different characteristics. Table 4 displays the albedo of the sampling line across the urban area of Beijing and its average, max, min, and variance in the business, high-density, and urban–rural mixed areas. However, the variance of the value is close to 0.002 which means that the maximum effect of cloud cover and other surfaces did not have much of an effect on the mean albedo.

**Table 4.** Seasonal albedo variation.

|                 |          | Spring | Summer | Autumn | Winter |
|-----------------|----------|--------|--------|--------|--------|
|                 | Avg      | 0.0941 | 0.1659 | 0.1102 | 0.0963 |
| Business        | Max      | 0.4698 | 0.482  | 0.4662 | 0.4431 |
|                 | Min      | 0.002  | 0.0205 | 0.0168 | 0.0312 |
|                 | Variance | 0.0025 | 0.0016 | 0.0021 | 0.0012 |
|                 | Avg      | 0.0972 | 0.1605 | 0.11   | 0.1075 |
| High-density    | Max      | 0.584  | 0.5378 | 0.6655 | 0.4126 |
|                 | Min      | 0.0048 | 0.0144 | 0.013  | 0.0348 |
|                 | Variance | 0.0014 | 0.0013 | 0.0013 | 0.0009 |
|                 | Avg      | 0.132  | 0.2004 | 0.1489 | 0.3875 |
| Urban–rural mix | Max      | 0.6629 | 0.5352 | 1.3168 | 0.1525 |
|                 | Min      | 0.0065 | 0.0247 | 0.0256 | 0.0466 |
|                 | Variance | 0.0016 | 0.0016 | 0.0018 | 0.0015 |

Figure 7 demonstrates the seasonal variation derived from Sentinel-2 in the business area. The most enormous albedo region in Figure 7 is in the center of the figures and all the values are nearly the same in spring and autumn. The spring (Figure 7A) and autumn (Figure 7C) albedo values are highly similar. The red circle's low albedo is in shadow areas resulting from the light blocking of the skyscrapers. The shadow area has high similarity in spring (Figure 7A) and autumn (Figure 7B) but becomes smallest in summer and largest in winter. The yellow circle is the high albedo region of the business area and winter's high albedo region is much smaller than that in the other three seasons.

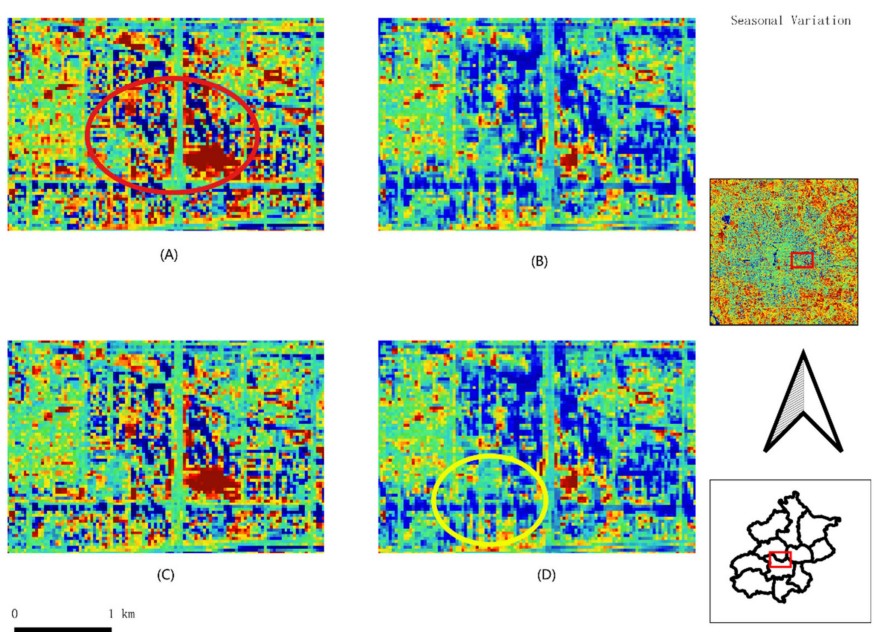

**Figure 7.** Seasonal variation in albedo derived from Sentinel-2 in business areas, Beijing (**A**) spring, 3 March 2020; (**B**) summer, 14 June 2019; (**C**) autumn, 19 September 2019; (**D**) winter, 14 December 2018.

Figure 8 indicates the seasonal variation derived from Sentinel-2 in the high-density area, which is very smooth in that all the albedos perform nearly 0.1 in spring, autumn, and winter. The spring (Figure 8A), autumn (Figure 8C) and winter (Figure 8D) albedo values are highly similar. Compared with other seasons, the characteristics of street albedo are not clear in the summer (Figure 8B). The street in the yellow circle represents where there is a medium-height building area. The red circle's area is the low-height building area which has minimal variation in spring, autumn, and winter.

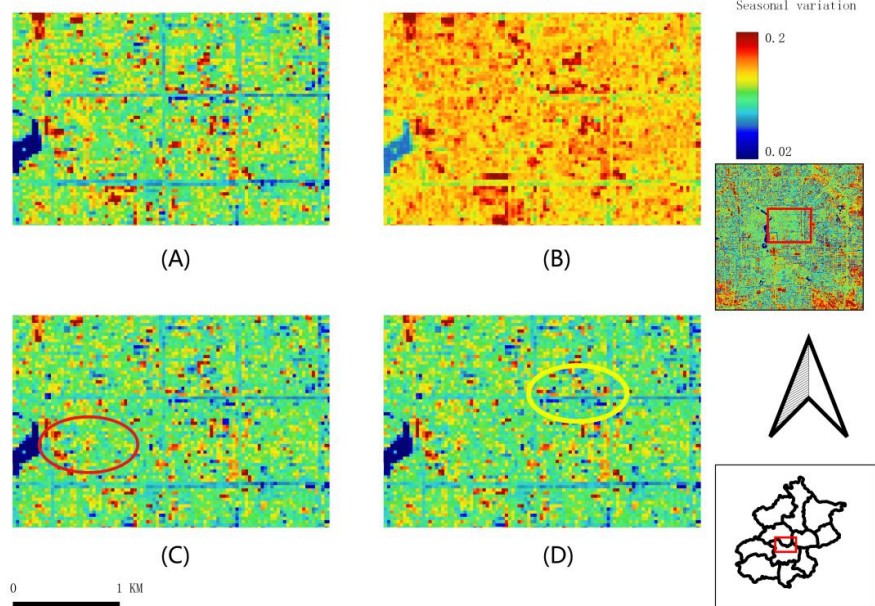

**Figure 8.** Seasonal variation in albedo derived from Sentinel-2 in high-density areas, Beijing (**A**) spring, 3 March 2020; (**B**) summer, 14 June; (**C**) autumn, 19 September 2019; (**D**) winter, 14 December 2018.

Figure 9 demonstrates the seasonal variation derived from Sentinel-2 in the urban–rural mixed area. The highest albedo value region is in the northeast of the region (blue circle in Figure 9A) with albedo close and over 0.2 in all seasons. The lowest albedo value region is on the southwest edge, where has large height difference. The yellow circle's albedo changed significantly in summer and autumn but was similar in spring and winter (yellow circle in Figure 9A).

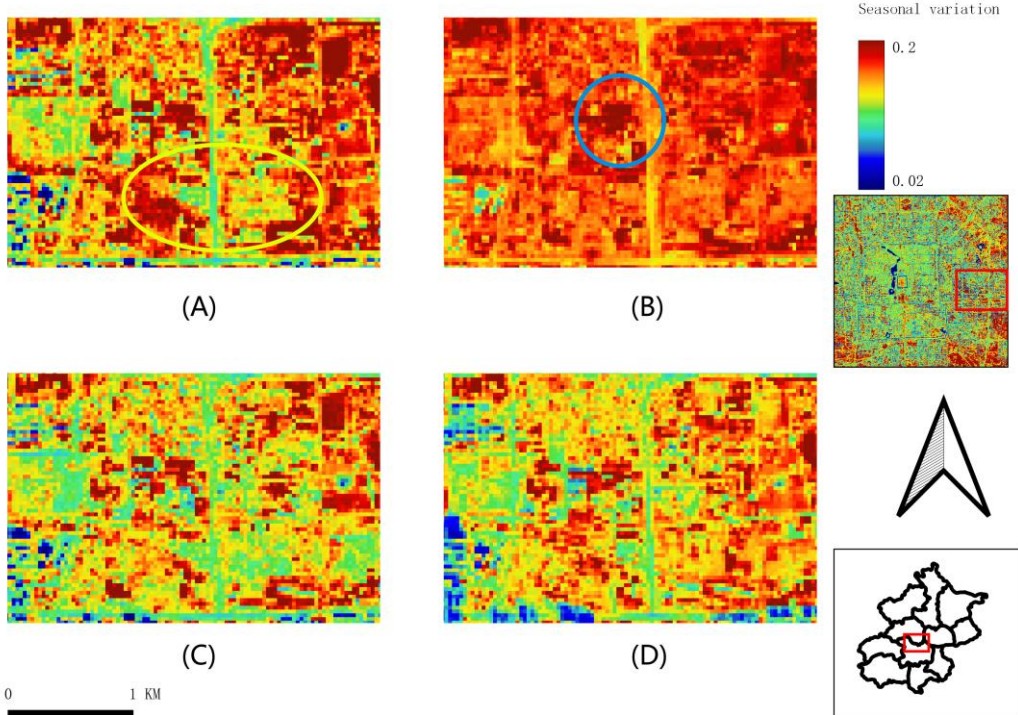

**Figure 9.** Seasonal variation in albedo derived from Sentinel-2 at mixed areas, Beijing (**A**) spring, 3 March 2020; (**B**) summer, 14 June 2019; (**C**) autumn, 19 September 2019; (**D**) winter, 14 December 2018.

Figure 10 shows the seasonal variation in albedo derived from Sentinel-2 along linear samples (Figure 1). This scatterplot is based on the linear sampling method. The y-axis (albedo) has a direct association with the change in the *x*-axis (longitude) due to the change in land cover and land use. The association between the albedo looks sensitive with the land cover change because there is unusually albedo in the typical land cover that changed long in the linear sampling region. The highest albedo of the center is the impervious surface square. The farmland and vegetation on the edge of eastern and western Beijing also have high albedos outside the city center (Figures 7–10).

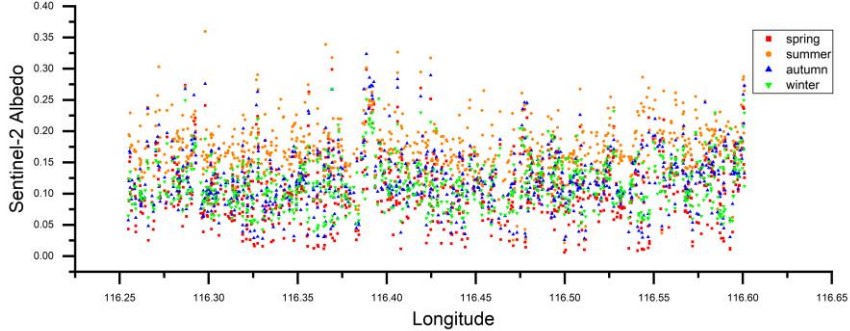

**Figure 10.** Seasonal variation in albedo derived from Sentinel-2 along linear samples.

Figure 11 displays a boxplot of seasonal variation in albedo derived from Sentinel-2 along linear samples. In the spring, the range of the albedo is larger than in other seasons. The summer variance of the albedo is the smallest and the 25th percentile to 75th percentile of albedo in summer is in a minimal range from 0.15 to 0.17. All significant *p* value of the samples between every two seasons are less than 0.01 which represents a significant difference.

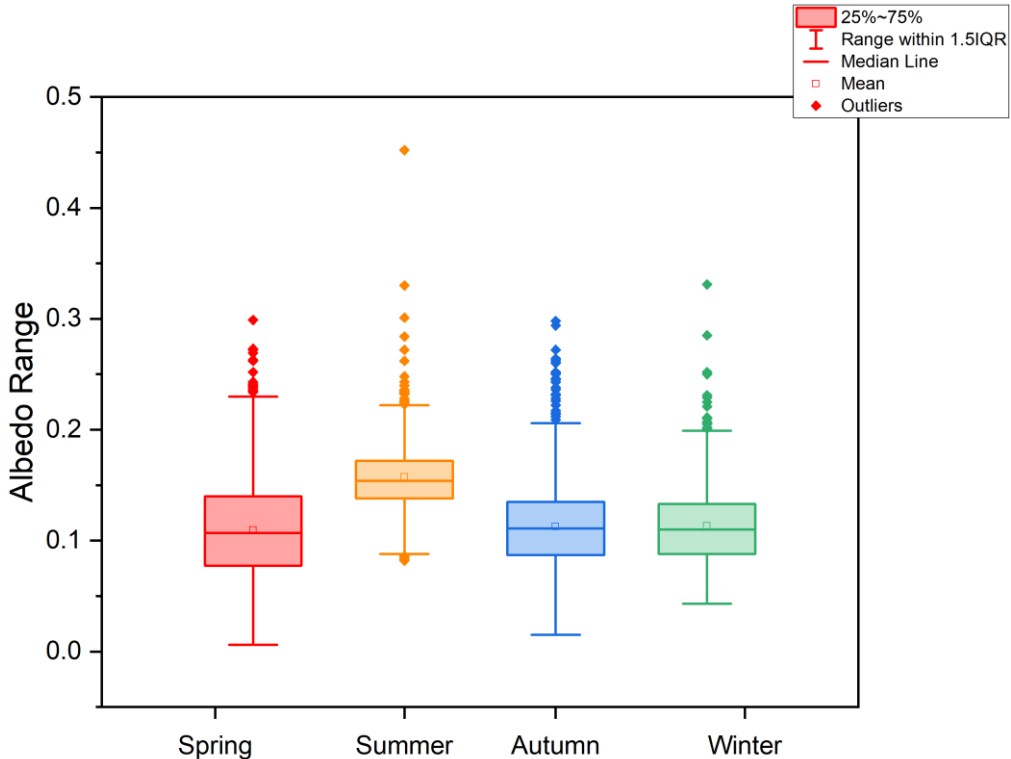

**Figure 11.** Boxplot of seasonal variation in albedo derived from Sentinel-2 along linear samples.

### 3.3. Relationship between Albedo and Other Indicators

Figure 12 is the boxplot of the Sentinel-2 albedo variation with buildings in summer and winter. The summer albedo reflection value is generally higher than that in winter. The low floors have the highest albedo in both summer and winter, and the albedo decreases slightly with increasing floors. The low floor albedo distribution range is largest in the summer, and the medium and high floor albedo values are almost the same in the summer. The median values of the albedos of medium and high floors are almost the same. However, the high floor range is higher than the medium floor range, which may be affected by the altitude variances among high buildings. The Sentinel-2 10-m resolution albedo data show that it can demonstrate the albedo variation with urban height (Figures 7–9). However, Figure 12 demonstrates no significant relationship between the urban building height and the albedo even in the winter. Some research [35,40] pointed out that bright materials and the canyon effect can neutralize one another's impacts on building heights [28,29,66,67].

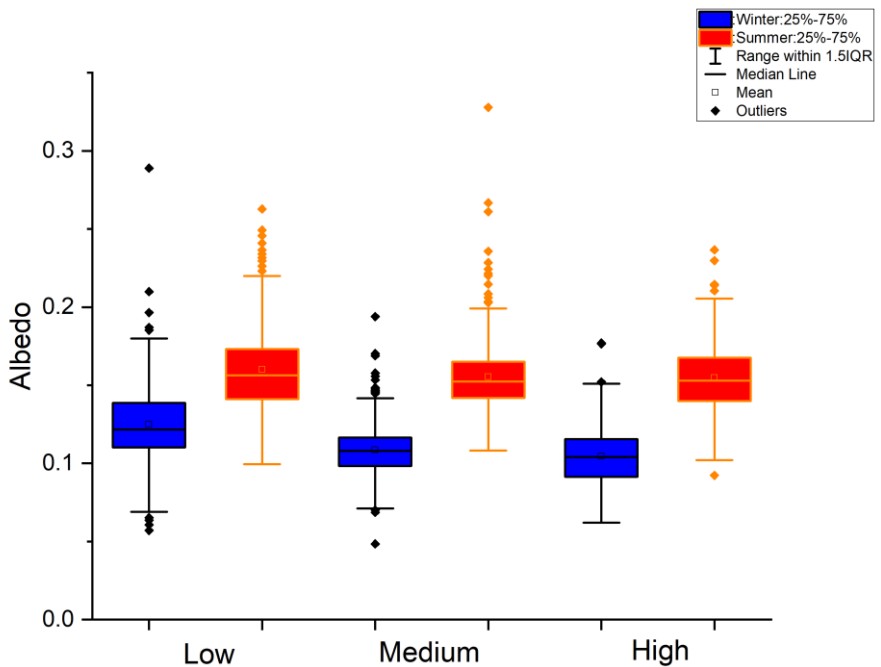

**Figure 12.** Boxplot of Sentinel-2 albedo variation with building height in Beijing in summer and winter.

Figure 13 displays the regression of segmented building density and albedo of Sentinel-2 sampling among Beijing in winter and summer. Figure 13 indicates a modem, negative, and linear association between the urban density and the albedo ($R^2$ = 0.5226 in winter and 0.25 in summer, $p < 0.001$). These significant regression relationships proved that the higher building density is, the smaller albedo will be in the winter.

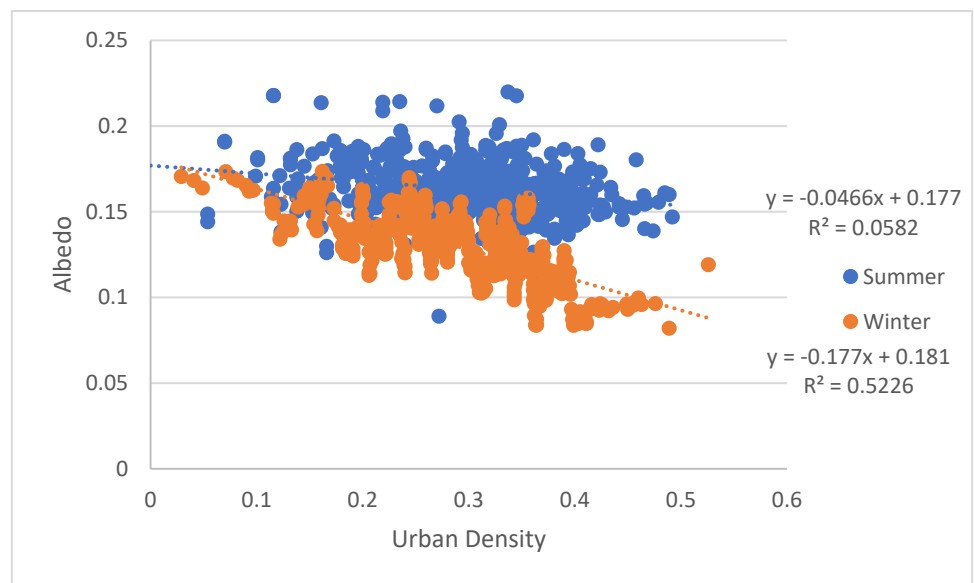

**Figure 13.** Regression of albedo and segmented building density in Beijing in winter (14 December 2018) and summer (14 June 2019).

Figure 14 shows the albedo variation in FVC sample points in Beijing in winter and summer. Beijing's FVC range is 0.01 to 0.63 in winter which is smaller than summer's FVC range from 0.01 to 0.92. The scatters of Figure 14 show a linear association between the albedo and FVC, but it shows a positive FVC–albedo relationship in winter ($R^2$ = 0.241, $p < 0.01$), but a very slightly negative relationship in summer ($R^2$ = 0.025, $p < 0.01$).

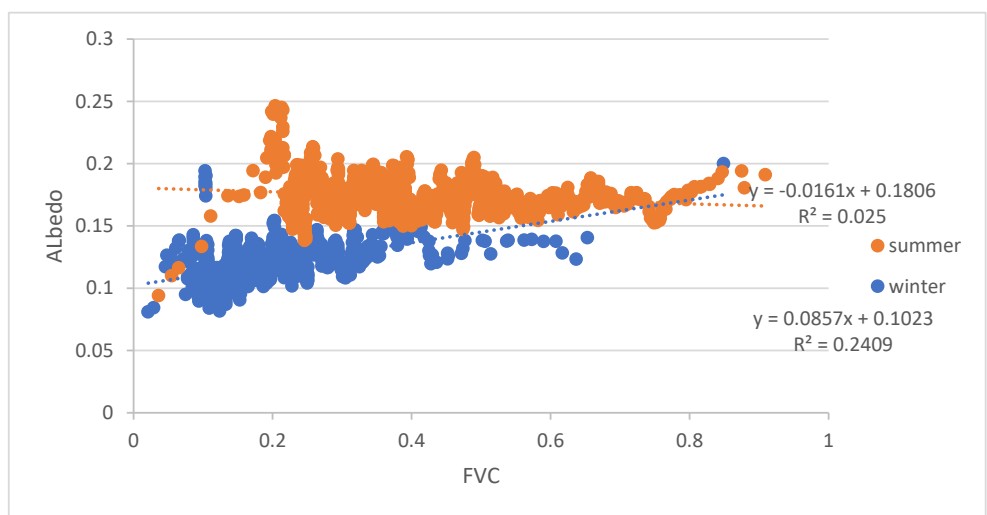

**Figure 14.** Regression of FVC and albedo in Beijing in winter (14 December 2018) and summer (14 June 2019).

Figure 15 displays the albedo variation of sample points with LST in Beijing in winter and summer. Beijing's winter LST is from 0 to 10 (B) and the summer LST's range is 27 to 45 (A). The figure proved a slightly positive linear association between the LST and the albedo and that albedo has a positive association with albedo in winter ($R^2 = 0.0236$, $p < 0.01$) and summer ($R^2 = 0.2367$, $p < 0.01$).

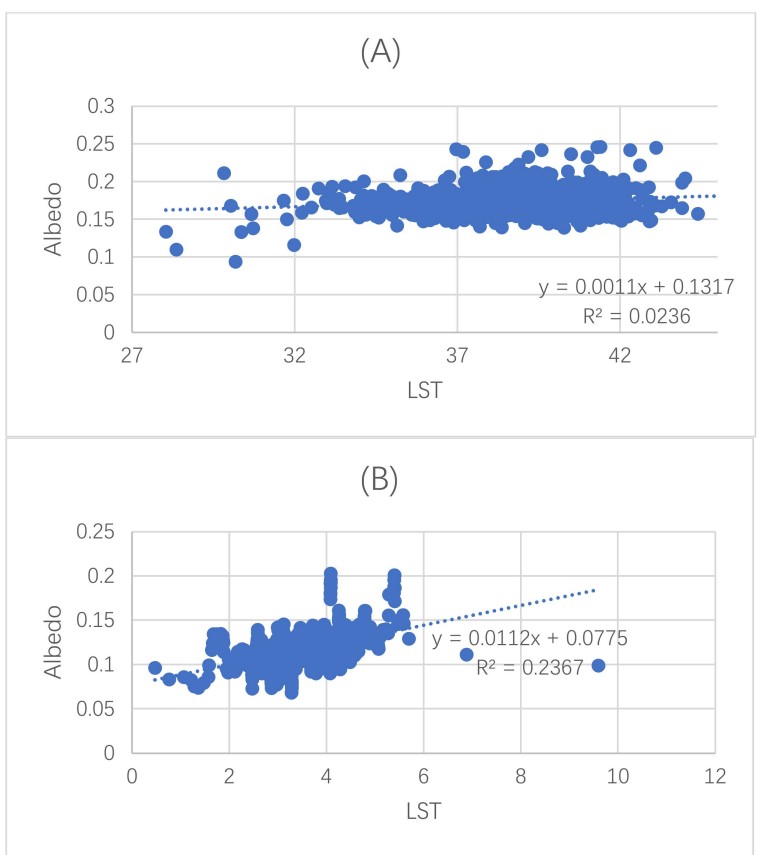

**Figure 15.** Regression of LST and albedo in Beijing: (**A**) in summer (14 June 2019) and (**B**) winter (14 December 2018).

*3.4. Albedo Association with Livability and Environmental Rating*

Table 5 indicates the average albedo, livability, and E-rating values in each of the four studied cities. Figures 16 and 17 show the variation in albedo among cities with different livability and environmental ratings (E-ratings). The table and boxplots indicate that the lower the albedo is, the higher the livability and the E-rating scores in the EIU report. Compared with the other three cities, Hong Kong and Tokyo have low albedo values and high livability and E-ratings. Bangkok has the largest albedo but the lowest livability and E-rating. The albedo effect exhibits a negative (Figures 16 and 17) relationship with livability and E-rating, but further study with data from more cities is needed.

**Table 5.** Albedo, LST, FCV, and livability and E-rating for Tokyo, Hong Kong, Beijing, and Bangkok.

| City | ALB | ALB Rank | Livability | E-Rating |
|---|---|---|---|---|
| Beijing | 0.0951 | 3 | 75 | 69 |
| Hongkong | 0.0836 | 1 | 91 | 83 |
| Tokyo | 0.0888 | 2 | 97 | 94 |
| Bangkok | 0.1151 | 4 | 66 | 63 |

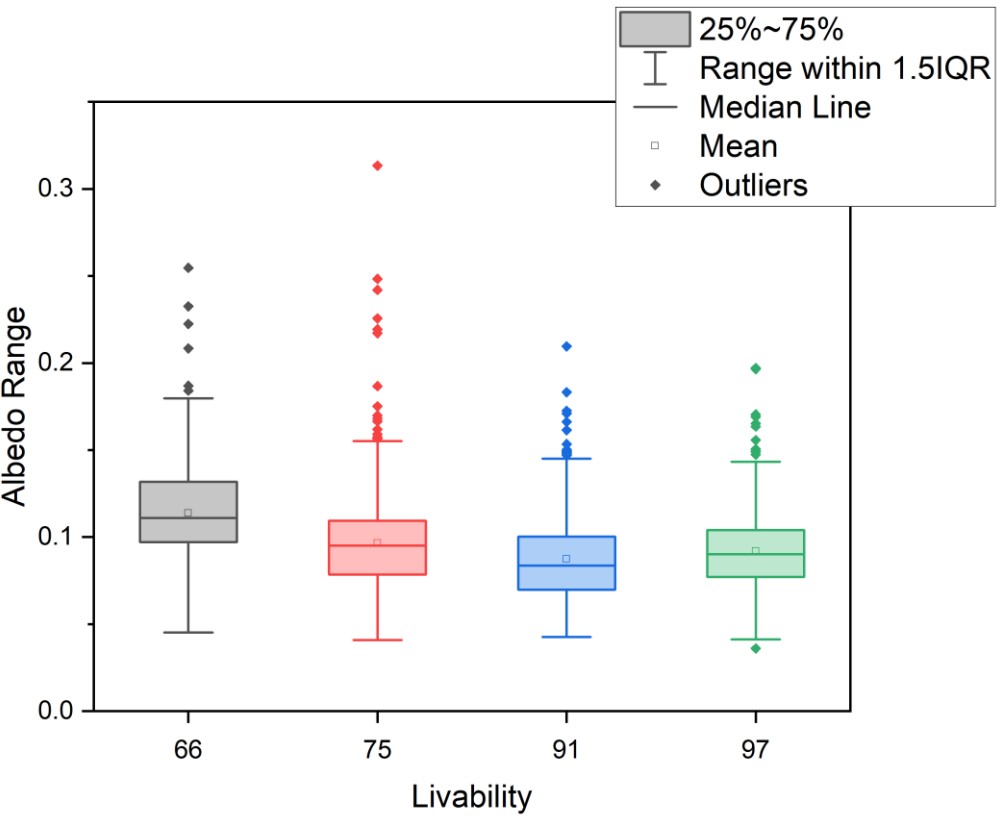

**Figure 16.** Boxplot of albedo ranges and livability.

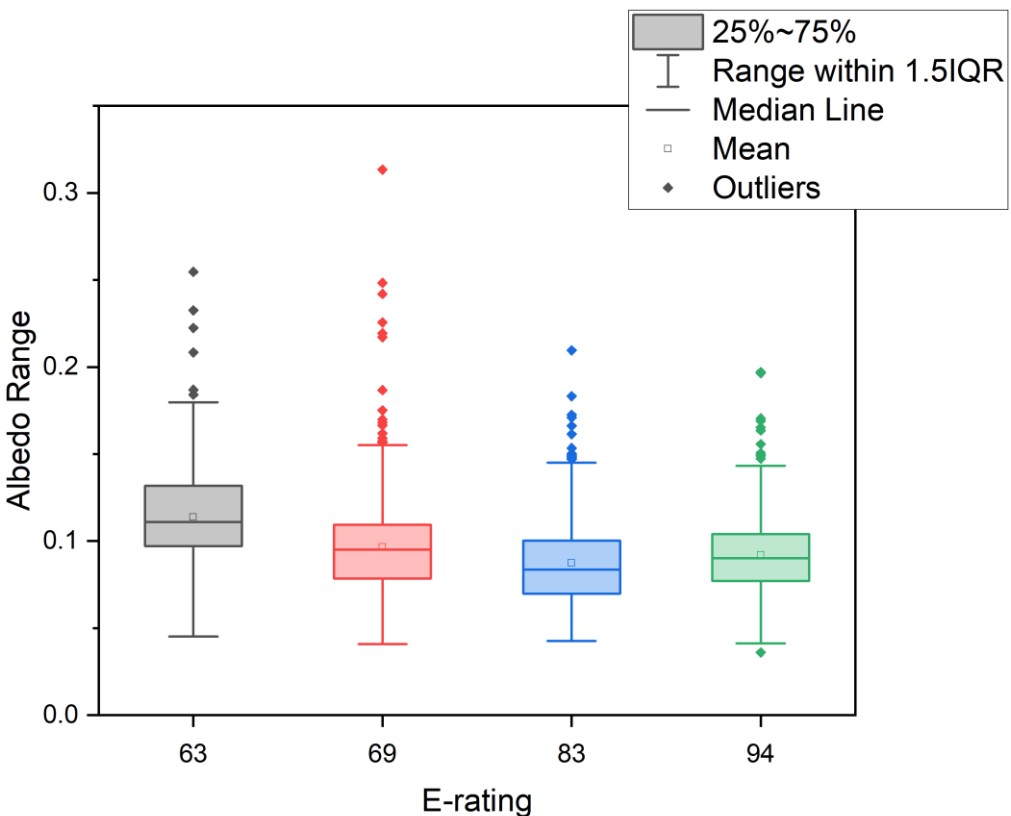

**Figure 17.** Boxplot of albedo ranges and environmental rating 4.

## 4. Discussion

Albedo has been widely used for estimating the variations in UHIs and glaciers [19–21,30,31]. This research has shown that urban surface albedo has high spatial heterogeneity (Figures 3–6) and seasonal variation (Figures 7–11), which could be captured by high-resolution data. In addition, this research demonstrates the association between albedo and other indicators (Figures 12–15). Thus, albedo can be considered a unique indicator for representing the attitude of an area's residents toward the urban environment as in glacier studies [17,41,52,68–70]. Furthermore, albedo can reflect livability and environmental rating (Figures 16 and 17) due to the variances in building materials and architectural styles in urban development [6,26,40].

The resolution has many effects on studies that relate albedo to other urban characteristics. Trlica's [26] research shows that albedo data at both 500 m and 30 m resolutions have no clear patterns representing land cover and land use. In this research, the albedo is not evenly distributed in the urban area due to light blocking caused by the height variation of buildings. Which was proven by Sugawara and Takamura [71] and Yang [24]. Guo's research [41] noted that various responses to albedo estimation among different cities could be explained by heterogeneity among the cities. The research demonstrates that heterogeneity exists not only among the cities but also in the internal environment of the city. The high-resolution albedo has a typical advantage in exploring the characteristics of the heterogeneity inside the different parts of the urban area (Figures 8–10).

This research selects different land cover and land use types for researching the albedo variation. The first is the mixed rural–urban area outside central Beijing with generally high albedo. The replacement of farmland and green cover by impervious land will significantly increase the albedo, a typical characteristic of albedo variation in urbanized areas [6,72]. The second is the high-density region displayed in Figure 4. Most of the high-density areas were built nearly a century ago in Beijing, and some parts have undergone reconstruction while other parts of it have not. This area has a significantly higher albedo than the business area due to its layout and materials, especially in winter. The third area is the business area

full of skyscrapers in Figure 3. Comparing the high-density urban area, the new layout of the street, building materials, and design have significantly reduced the albedo. On the other hand, the canyon effect caused by the altitude variances further reduce the albedo, which is proven by Jacobson [35] and Trlica [26].

The research contents indicate that the Sentinel-2 (10 m resolution) albedo data have much higher precision than Landsat 8 (30 m resolution) data in regions with low albedo values (Figures 3–5). The output indicates that Sentinel-2 albedo data recognized water, public green spaces, and building canyon effect regions that the Landsat 8 data could not distinguish as Figure 7. Previous studies [73] also found that the albedo characteristics of building canyons can affect their surrounding areas and stack the albedo effects. The lower the resolution of the image data is, the higher the stacking effect of the albedo canyon effect. This performance might be generated from the adjacency of the remote sensing data. Multiple studies emphasize that the adjacency effect can affect reflection [74–76]. Although the data in this research have been atmospherically corrected, the adjacency effects still occur on the water surface and in green spaces. [74]. The albedo of Sentinel-2 shows that the adjacency effect should be considered in medium and low-resolution data, and in 30-m resolution albedo data (Figures 3 and 4). However, high-resolution data (10-m resolution) can significantly reduce the influence of the adjacency effect and raise the accuracy of the remote sensing albedo mapping and statistics. Therefore, the higher the resolution of remote sensing data is, the more accurate the measurement of the canyon effect will be, emphasizing that each component in the urban landscape displays unique characteristics of radiant heat, humidity, and aerodynamics [73,77].

On the other hand, there is also a seasonal heterogeneity in the albedo in the urban area. Previous researchers have focused on the annual variation in albedo [2,17,26,39] instead of the seasonal variation due to the limited temporal resolution of remote sensing [26,27]. This research proved that seasonal variation mapping of the albedo helps highlight some city characteristics. In Figure 11, most regions display the characteristics of high albedo in summer and low albedo in winter. We find that the variance in the albedo value is the smallest in summer and largest in winter. The albedo distribution in spring and autumn are relatively different. In addition, the median value of albedo in summer is the largest, while that in other seasons is almost unchanged and sampling is very dispersed at the same time. This performance can also be confirmed by the summer albedo performance of Figures 8–10. It is highly possible that the summer albedo distribution is affected by the zenith angle when analyzed with the angle data of the sensors Table 6. The albedo values of areas with no altitude variances, such as squares, are incredibly high and close during all four seasons. In areas with medium and high urban densities, the canyon effect caused by building altitude differences will induce a lower albedo value in summer. However, the albedo is not stable due to the occlusion effect of buildings with seasonal variation which results in different effects on the albedo value within the same area depending on the season.

**Table 6.** Mean sensor angles of the Sentinel-2 Images.

|  | Spring | Summer | Autumn | Winter |
|---|---|---|---|---|
| MEAN AZIMUTH ANGLE | 157.0893 | 142.9314 | 163.9391 | 166.7776 |
| MEAN ZENITH ANGLE | 49.48056 | 21.47778 | 45.88404 | 64.49991 |

The high-resolution albedo variation of seasonal variation and resolution demonstrates that the urban layout and the building height can affect the surroundings' albedo value. Previous research [78] explained that the canyon effect acts on impervious surfaces by the altitude differences of buildings. This research applied sampling data to verify whether the urban layout and building height are directly related to the albedo effect. Figures 12 and 13 proved that there are associations between the urban density and floor height in Beijing and those associations can be modified with the seasons in Figure 11.

Previous research claimed that albedo has a relationship with FVC [79–81] and LST [32,82,83]. However, we found weak associations between the albedo and LST and FVC (Figures 14 and 15). This phenomenon could be caused by the canyon and building materials to the albedo, while other factors, such as temperature, impervious surface materials, and human activities can influence the LST. FVC is associated with urban expansion and seasonal variation. There is no absolute causation between the LST, FVC, and albedo [33], although they overlap in some dynamics.

The albedo variation displays the heterogeneity inside Beijing where different sample regions have different heterogeneity. It is difficult to collect livability and environmental rating data in a specific area of the city. Therefore, this research examines the albedo value in different cities to examine the albedo's impacts. Figures 16 and 17 display the relationship with livability and E-rating. Enete's research explained that [73] high livability and E-rating cities have more motivation to regular building materials and layout distributions to control the effect of albedo on the UHI and urban thermal environment to achieve better livability and living environment ratings. The variation of albedo and livability, E-rating ranking (Figures 16 and 17) proved that the albedo might reach a minimum value with the development stage of an urban area in a certain period of rapid urban development. Albedo is recognized as a critical parameter for controlling urban temperature [19–21,30,31,35] and urban decision-makers tend to redevelop urban areas. The redevelopment of urban areas will release the heat from urban area by expanding street-wide, increasing the number of parks, and applying new materials to reduce light energy absorption and the canyon effect for a better livable environment. However, it is difficult to confirm that the albedo and LST are very related when the samples are used in a correlation analysis (Figure 15) in this research. Some studies [20,33] stated that the albedo regions have cooler air temperatures by increasing the albedo. These phenomena are consistent with repairing imperviousness in urban development in the face of changes in the natural environment, destruction, and redevelopment for livability. Therefore, albedo can describe not only the spatial heterogeneity of urban building materials and layouts but also the temporal heterogeneity that can be used to describe different stages of urban development.

Although albedo in this study has proven that albedo can be an efficient indicator for representing the spatial heterogeneity and temporal variation in urban areas, there are some limitations that remain. Current research is mainly focused on the characteristics of land cover, urban morphology, and urban form's impact on albedo, while the impact of urban thermal properties on albedo should also be considered. The complex urban geometry [84], building materials [24], and absorption and emission rates of different surface materials also affect albedo. On the other hand, the association between albedo and urban environment and livability also needs to be explored including urban thermal comfort [85], and anthropogenic heat [86].

## 5. Conclusions

Albedo has been altered due to urban expansion, which significantly impacts local and regional climates. This research demonstrates that albedo is a unique indicator in urban environment research. The study shows that the albedo contains significant spatial heterogeneity and seasonal variation on the surface of the urban areas. Urban surface albedo at high spatial resolution can better represent spatial-temporal variation, building canyons, and pixel adjacency effects. The albedo of the urban surface is associated with building density and height, land surface temperature (LST), and fractional vegetation cover (FVC). Resolution and seasonal variation sampling demonstrate that albedo is associated with urban land cover and the stage of development of that area. Further analysis confirms that the development of the urban area can impact the surface albedo, which is indicated by the EHI livability and environment rating albedo of the urban surface high-resolution data can improve the acknowledgment of urban environment and human livability with wider application in the urban environmental research.

**Author Contributions:** This paper is conceptualized by H.W. and B.H.; H.W. conducted the main experiment and prepared the original manuscript; B.H., Y.Z., Z.Z. and Z.M. reviewed the manuscript. All authors have read and agreed to the published version of the manuscript.

**Funding:** This work was supported by the Open Fund of State Key Laboratory of Remote Sensing Science, Aerospace Information Research Institute, Chinese Academy of Sciences (Grant No. PFSLRSS202023).

**Data Availability Statement:** Landsat-8 and Sentinel-2 data are available from GEE. Urban Block data are available from Beijing City Lab. Livability and Environmental-rating data are available from Economics Intelligence Unit website.

**Conflicts of Interest:** The authors declare no conflict of interest.

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
