# Peer review of "Spatial Heterogeneity and Temporal Variation in Urban Surface Albedo Detected by High-Resolution Satellite Data"

_remotesensing, doi:10.3390/rs14236166_

Round 1
Author Response
Please see the attachment。

Reviewer 2 Report
Dear Authors,
In general, I've enjoyed reading this paper and in my opinion this is a good one. There are few questions/point I'd like to add:
Line 141: last sentence, "Even though the expansion of Tokyo is almost over, its rede-140 velopment is continuing.3. Results" - what does '.3. Results' states for - is this typo. If not could you rephrase it so it could be cleared to the reader.
Line 161: "..which can not one hundred percent remove all clouds and shadows." -> 'which can not remove, one hundred percent, all clouds and shadows'
Line 328: "This area’s albedos are much different with different sensors, albedo of Landsat-8 is about 0.1" - can you suggest what differences in sensors can lead to that. (this is just suggestion; If you can not find suitable answer, please ignore this)
Line 401-402: "The villa area’s.." - what does 'villa' points to. Looking from the perspective of unfamiliar reader, could you make it more clear.
Line 541: "The output indicates that Sentinel-2 albedo data recognized water, public green spaces and building canyon effect regions that the Landsat 8 data could not distinguish as" - could you suggest why this is the reason? Also this sentence ends with 'as Error! Reference source not found..' I've assume this is error while converting paper for review. If there is some text missing in this review version which explains question, than please ignore this suggestion.
I was a pleasure reviewing this paper!
Keep up good work
Regards
Reviewer 3 Report
Comments on the paper entitled on the “Spatial heterogeneity and temporal variation of urban surface’s albedo detected by high-resolution satellite data” submitted to the ‘Remote Sensing’. The paper mainly deals with the spatial temporal variation of the urban surface albedo using high resolution data. There some issues that need to be addressed before publication the research work.
1. The abstract needs to be revised. Where this study was carried out? It should be mention in the abstract. The research objectives are not clear in the abstract.
2. In the introduction section, I missed the various sources through which Albedo or surface albedo can be detected. I would suggest to incorporate few literatures in the introduction section:
https://doi.org/10.1016/j.jclepro.2022.130804.
https://doi.org/10.1016/j.isprsjprs.2022.02.008.
https://doi.org/10.1016/j.rse.2021.112321.
https://doi.org/10.3390/rs14010143.
https://doi.org/10.1016/j.rse.2020.111980
3. The data source should be presented in table.
4. What does line 182 mean? Clarify it.
5. Box plot is a statistical analysis or graphical analysis (section 2.3.5)
6. The results should be concised.
7. A comparison should be made between the datasets used in this study (sentinel and landsat)
8. What are the limitations of the study.
